# A Meta-Analysis of Comparative Transcriptomic Data Reveals a Set of Key Genes Involved in the Tolerance to Abiotic Stresses in Rice

**DOI:** 10.3390/ijms20225662

**Published:** 2019-11-12

**Authors:** Matteo Buti, Elena Baldoni, Elide Formentin, Justyna Milc, Giovanna Frugis, Fiorella Lo Schiavo, Annamaria Genga, Enrico Francia

**Affiliations:** 1Department of Life Sciences, Centre BIOGEST-SITEIA, University of Modena and Reggio Emilia, Via Amendola 2, 42124 Reggio Emilia, Italy; matteo.buti@unifi.it (M.B.); justynaanna.milc@unimore.it (J.M.); enrico.francia@unimore.it (E.F.); 2National Research Council (CNR), Institute of Agricultural Biology and Biotechnology (IBBA), Via Bassini 15, 20133 Milano, Italy; annamaria.genga@ibba.cnr.it; 3CNR-IBBA, Rome Unit, via Salaria Km. 29,300, 00015 Monterotondo Scalo (Roma), Italy; giovanna.frugis@cnr.it; 4Department of Biology, University of Padova, 35131 Padova, Italy; elide.formentin@unipd.it (E.F.); fiorella.loschiavo@unipd.it (F.L.S.); 5Botanical Garden, University of Padova, 35123 Padova, Italy; 6Present address: Department of Agriculture, Food, Environment and Forestry, University of Florence, 50144 Florence, Italy

**Keywords:** *Oryza sativa* L., abiotic stress, meta-analysis, comparative transcriptomics, gene co-expression network, abscisic acid, jasmonic acid, transcription factors, tolerance genes

## Abstract

Several environmental factors, such as drought, salinity, and extreme temperatures, negatively affect plant growth and development, which leads to yield losses. The tolerance or sensitivity to abiotic stressors are the expression of a complex machinery involving molecular, biochemical, and physiological mechanisms. Here, a meta-analysis on previously published RNA-Seq data was performed to identify the genes conferring tolerance to chilling, osmotic, and salt stresses, by comparing the transcriptomic changes between tolerant and susceptible rice genotypes. Several genes encoding transcription factors (TFs) were identified, suggesting that abiotic stress tolerance involves upstream regulatory pathways. A gene co-expression network defined the metabolic and signalling pathways with a prominent role in the differentiation between tolerance and susceptibility: (i) the regulation of endogenous abscisic acid (ABA) levels, through the modulation of genes that are related to its biosynthesis/catabolism, (ii) the signalling pathways mediated by ABA and jasmonic acid, (iii) the activity of the “Drought and Salt Tolerance” TF, involved in the negative regulation of stomatal closure, and (iv) the regulation of flavonoid biosynthesis by specific MYB TFs. The identified genes represent putative key players for conferring tolerance to a broad range of abiotic stresses in rice; a fine-tuning of their expression seems to be crucial for rice plants to cope with environmental cues.

## 1. Introduction

Environmental stresses are the most critical factors that affect crop growth and development, causing plant damages and injuries that may lead to yield loss [1]. During their life cycle, plants are frequently affected by several stresses that cause general or specific effects on growth and development [2]. When plants are exposed to abiotic constraints, the perception and transduction of stress signals induce the plant to activate stress-related genes, resulting in metabolic and physiological changes that adapt the organism to the new environmental conditions [2,3]. The ability to tolerate abiotic stresses drastically varies among different genotypes within the same species. Studies concerning the regulatory networks behind stress response are essential for identifying the genes that are involved in tolerance and developing potential applications for crop improvement [4,5,6,7,8]. Increased knowledge regarding the molecular mechanisms of adaptation to environmental constraints, which leads to the development of new resilient varieties, still represents the best strategy for coping with the erratic nature of stress caused by global change.

It is well known that plant responses to different abiotic stresses share many regulatory mechanisms, and intensive cross-talk among different signalling pathways occurs [9,10]. In particular, several investigations studied the shared response to cold, drought, and salt stress in plants, evidencing the central role of phytohormones in the cross-talk among different signalling pathways [11,12,13,14]. The role of abscisic acid (ABA) in abiotic stress response is well-known and extensively described [10], and recent research has revealed new insights into the mode of action of jasmonic acid (JA) and ethylene in plant abiotic stress tolerance [15]. JA and its derivatives, the jasmonates, are known to regulate stress-responsive and developmental processes [16]. The cross-talk between different plant hormones results in synergistic or antagonistic interactions that play crucial roles in plant response to abiotic stresses. Cross-talk among ABA, JA, and ethylene-mediated signalling pathways in response to different abiotic stresses has been extensively reported [14,17,18]. Moreover, different abiotic stresses, such as drought, salt, and cold stress, have, as a common consequence, the accumulation of reactive oxygen species (ROS) [19]. Higher plants have evolved in the presence of ROS and they have acquired dedicated pathways to protect themselves from ROS toxicity, as well as to use ROS as signalling molecules [19]. Interestingly, the ROS signalling pathway interacts with phytohormones, especially with ABA, to regulate the protective responses of plants against biotic and abiotic stresses [20,21]. 

Rice (*Oryza sativa* L.) is one of the most important crops worldwide, since it represents a staple food for more than half of the world population [22]. Abiotic stresses, such as low temperatures, drought, and soil salinity, are among the major factors limiting its productivity [23,24]. For this reason, a primary goal of rice breeding programs is the development of genotypes that are able to tolerate adverse environmental conditions. Recent advances in RNA sequencing techniques have provided highly efficient and relatively low-cost methods to analyse whole transcriptomes, allowing for a full characterization of the transcriptomic response of organisms to external stimuli. These new technologies have been used in several studies to characterize the genetic background of stress-tolerant rice varieties, while comparing the transcriptomic response of tolerant and sensitive genotypes to single stresses, in particular cold [5,25,26,27], drought [28,29,30,31], and salt stress [32,33,34,35,36]. These analyses allowed for the identification of genes and pathways that play major roles in conferring tolerance to single environmental cues. Comparative investigation of gene expression networks in rice genotypes might also be applied to study the response to different abiotic stresses and identify master players orchestrating the complex mechanisms of stress response. However, very few comparative studies were reported, aiming at the identification of genes that confer tolerance to more than one stress. To our knowledge, only two studies are present in the literature exploring the response to salt and drought stresses in rice, through a comparative transcriptomic analysis of tolerant and sensitive genotypes [37,38]. The use of meta-analyses of transcriptomic data is strongly increasing to dissect the regulatory networks of plant-environment interactions in many plant species; this approach provides robust and meaningful information, even with a limited number of datasets [39,40,41,42,43]. A meta-analysis on drought stress response identified conserved drought-adaptive genes that are shared across different species [40]. Recently, Smita et al. [43] published a meta-analysis with a broad approach and identified gene network modules that are associated with abiotic stress tolerance in rice. These kinds of investigations may have applications in crop genetic improvement for achieving tolerance to multiple stress conditions.

In the present paper, raw transcriptomic data of tolerant and susceptible rice genotypes in response to chilling [25], osmotic [28], and salt [32] stresses were re-analysed with a unique bioinformatic pipeline to identify the differentially expressed genes (DEGs) for each stress condition. The resulting data were then integrated to identify those genes that seem to be involved in the differentiation between tolerance and susceptibility to all of the three stress conditions. Moreover, a Gene Co-expression Network (GCN) was constructed to identify the related signalling and metabolic pathways. The genes that represent the hubs of the network are putative candidates for improving tolerance to a broad range of abiotic stresses in rice. The physiological contribution to stress tolerance of these genes, in the frame of the identified signalling and metabolic pathways, is compared with literature data and extensively discussed.

## 2. Results and Discussion

This study aims to identify the genes that may represent putative common key players for conferring tolerance to abiotic stresses in rice, as well as the signalling or metabolic pathways in which these genes are involved. RNA-sequencing data of rice seedlings grown under control and stress conditions were previously generated to perform three independent comparative analyses of the response to chilling, osmotic, and salt stresses of selected tolerant and susceptible rice genotypes [25,28,32]. For each stress experiment, a phenotypic analysis of several cultivars was performed and a couple of contrasting genotypes were selected for the subsequent transcriptomic analysis, namely the two genotypes showing the most contrasting responses (tolerance vs. susceptibility) to that specific stress condition [25,28,32]. For this reason, the three couples of contrasting genotypes selected for chilling, osmotic, and salt stress experiments were different (see Table 1). This is an advantage for the meta-analysis here presented, since, for each experimental condition, two genotypes that highly differ in their physiological response to stress are available. Differently, a unique couple of genotypes could not have shown highly contrasting phenotypes for all the three stress treatments. Moreover, the use of different genotypes allowed to avoid genetic bias that might derive from using the same two rice genotypes for all of the experiments. In the present meta-analysis, we separately re-analysed the three transcriptomic data with a unique bioinformatic pipeline to identify DEGs that were related to each stress condition. Subsequently, we integrated the results of the three experiments to select the genes whose stress-induced expression variation differed between the tolerant and susceptible genotypes under all three stress conditions. Figure 1 illustrates the overall work-flow, from data mining to candidate genes identification.

### 2.1. RNA-Seq Data Processing

The raw RNA-Seq data of the samples described in Table 1 were processed, as described in Materials and Methods. Read counts were generated after quality filtering and mapping on the rice reference genome. After assessing the high quality of the RNA reads, the adapters and low-quality nucleotides were filtered out, resulting in a percentage of “survived” reads of 97–98% for salt and osmotic stresses and of 91–93% for chilling stress (Appendix A). The observed difference was due to the sequencing approaches used (i.e., single-end for salt and osmotic stress experiments vs. paired-end for chilling stress experiment, see Table 1). Indeed, with the paired-end reads sequencing approach, a sequence survives to filtering only if both reads are not filtered out, thus resulting in a lower percentage of filtered reads. For each RNA library, 94–97% of filtered reads were aligned to *Oryza sativa* ssp. *Japonica* reference genome (Appendix A), and, for each RNA library, the number of reads mapping to each predicted rice gene was calculated. The normalization factors were calculated according to library sizes (Appendix A), and expression values of all active genes were reported as normalized read counts on Appendix A. Multi-dimensional scaling (MDS) plots for the RNA samples showed high reproducibility among biological replicates, as the replicates were all closely clustered and clearly separated from the other samples (Appendix A).

### 2.2. Differential Gene Expression in Response to Each Single Stress

Normalized read counts were used as input data for differential expression analysis in the treated vs. control plants. The results are reported in Appendix A and graphically represented in Appendix A. As shown in Table 2, a comparable number of DEGs was found in chilling (14,944 and 14,355 DEGs for susceptible and tolerant genotypes, respectively) and osmotic stress experiments (13,722 and 12,654 DEGs for susceptible and tolerant genotypes, respectively), while, in the salt experiment, the number of DEGs was lower (6197 and 1537 DEGs for the susceptible and tolerant genotypes, respectively). 

This discrepancy is probably due to the different timing of sampling in the three experiments. Indeed, the amplitude of the activated transcriptome reflects the phases of the stress response: in the earliest events of stress signalling, a large-scale transcriptional response is activated, whereas, in the later phases of stress, the amplitude of the transcriptional response becomes lower [44]. The duration of chilling and osmotic treatments that were considered for the current analyses reflected a similar phase of stress response dynamics. In leaves, the number of DEGs was higher after 10 h than after 2 h of chilling stress [25] and, similarly, after 24 h than after 3 h of osmotic stress [28]. Differently, for salt treatment, a later phase of the stress (72 h), when the amplitude of the transcriptional response is probably lowering, was considered to evaluate the response to the ionic component of salt stress [32]. High salinity imposes both osmotic and ionic stress on plants (Munns and Tester, 2008). Osmotic stress is established early after the stress onset, while ionic stress depends on ion accumulation in aerial parts of the plant and, thus, it takes a longer time to be perceived by plant cells [6,45]. 

The number of DEGs was similar between the susceptible and tolerant genotypes in chilling and osmotic stress experiments while, for salt stress treatment, the number of DEGs was four times higher for the susceptible genotype when compared to the tolerant genotype. This difference can be ascribed to a delay of the response of the salt-susceptible genotype. Consistently, Formentin and colleagues [46,47] demonstrated that the salt-tolerant genotype is able to activate a faster stress response as compared to the salt-susceptible genotype, through an early activation of H_2_O_2_ and hormone signalling pathways. 

A comparison of the extent of transcriptomic responses to the three stresses in tolerant and susceptible genotypes was carried out intersecting the six groups of DEGs related to the different treatments and genotypes (SaTol, SaSus, OsTol, OsSus, ChTol, and ChSus); among the 61 possible intersections between the six groups of DEGs, the 30 intersections with the highest number of DEGs were visualized, as shown in Figure 2. 

In particular, the analyses highlighted that 186 and 84 genes were up- or down-regulated, respectively, in all three stress conditions and in all the tolerant and susceptible genotypes (highlighted in red in Figure 2, Appendix A). These genes can be considered as being broadly involved in the response to abiotic stresses, regardless of the type of stress and response phenotype (tolerance or susceptibility). The GO-enrichment analysis underlined that the 186 up-regulated genes resulted in being principally enriched for the categories “response to stress” (GO:0006950), “cellular nitrogen compound metabolic process” (GO:0034641), “cofactor binding” (GO:0048037), and “response to stimulus” (GO:0050896) (Appendix A). Differently, the GO-enrichment analysis of the 84 down-regulated genes gave no significantly enriched terms, suggesting that no particular class of genes decreasing their activity was represented. The low number of common DEGs was probably due to the difference in stress duration among the three experiments. Consistently, chilling and osmotic stress experiments, whose durations were similar, shared a relatively higher number of common up- or down-regulated genes (1554 and 1173, respectively; Figure 2).

### 2.3. Identification of Genes Differentiating the Response of Tolerant and Susceptible Genotypes 

Starting from DEGs that were derived from each experiment (Table 2, Appendix A), the stress-induced variation in gene expression, observed in tolerant and susceptible genotypes, was compared, and we considered six classes of interest: (i) genes that were up-regulated only in the susceptible genotype and unchanged in the tolerant one (“SusOnly_up”), (ii) genes that were down-regulated only in the susceptible genotype and unchanged in the tolerant one (“SusOnly_down”), (iii) genes that were up-regulated only in the tolerant genotype and unchanged in the susceptible one (“TolOnly_up”), (iv) genes that were only down-regulated in the tolerant genotype and unchanged in the susceptible one (“TolOnly_down”), (v) DEGs in both genotypes, showing a difference of LFC (ΔLFC = LFC of tolerant sample—LFC of susceptible sample) between tolerant and susceptible cultivars higher than 1 (“ΔLFC > 1”), and (vi) DEGs in both genotypes, showing a difference of LFC between tolerant and susceptible cultivars less than -1 (“ΔLFC < −1”). Therefore, for each stress experiment, genes that were listed in classes (i), (ii), (iii), or (iv) represented those that were only up- or down-regulated in one genotype (tolerant or susceptible) and unchanged in the other genotype. Differently, genes that were listed in classes (v) or (vi) represented those genes that were differentially expressed in both tolerant and susceptible genotypes for each stress condition, either with a similar (up-up or down-down) or a contrasting (up-down or down-up) modulation, with a difference between the LFC values of the two cultivars >1 or <−1. Appendix A provides a list of the selected genes for each stress. In particular, 5898, 7066, and 5640 genes were listed in the six classes for chilling, osmotic, and salt stresses, respectively (Table 3). 

We assumed that the expression regulation of such genes would differ within each couple of genotypes (tolerant vs. susceptible), after the occurrence of all of the three considered stresses, in order to identify the genes that may represent common key players for conferring tolerance to chilling, osmotic and salt stresses in rice. Subsequently, we focused our interest on the 420 genes that are at the intersection among the three data sets, as shown in Figure 3. These are the genes that fulfilled one or another of our selection criteria in all of the three stress conditions, thus being common to all of them. Therefore, they were considered as potential candidate genes that differentiate the responses of tolerant and susceptible genotypes under the three environmental constraints (Figure 3a, Appendix A).

The criterion we used to define these genes as potential candidates was their different behaviour between the tolerant and the susceptible genotypes under each of the three stresses, regardless of whether or not the same trend of regulation (up or down) was shown in all three stresses, based on the consideration that the mechanisms that are involved in sensing and signalling the three stresses are different. GO-enrichment analysis of these 420 genes (Figure 3b, Appendix A) revealed the enrichment of the classes “response to stress” and “response to stimulus”, as expected. Moreover, several enriched classes were related to gene transcription (e.g., “regulation of transcription, DNA-dependent”, “transcription regulator activity”, “transcription factor activity”, “RNA biosynthetic process”), which suggests a major role of TFs in reconfiguring the transcriptome profile in abiotic stress responses. Among these 420 genes, we searched for the putative master regulators conferring stress tolerance and the pathways in which they are possibly involved.

### 2.4. Characterization of Genes and Pathways Putatively Involved in Abiotic Stress Tolerance

We performed a pairwise correlation analysis of all pairs of these 420 genes (Appendix A) and constructed a Gene Co-expression Network (GCN) using the Cytoscape platform to determine the relationships among the 420 selected genes and possible common pathways that putatively differentiate the response of tolerant and susceptible genotypes [48]. In a GCN, nodes represent genes, and edges connect the nodes if the corresponding genes are significantly co-expressed across the samples [49]. Network topology is defined as the layout of nodes and edges, and the topological properties determine the functional aspects of the relationships [49]. Parameters that are derived from network local properties, such as clustering coefficient, node degree (number of connected nodes), betweenness, and closeness centrality, are commonly used for node ranking [48]. Nodes with a higher rank (i.e., with a high degree of connection and high betweenness centrality) are identified as major hubs, and they are likely associated to essential genes in the network [48]. 

Using absolute Pearson’s correlation (|r|) values ≥ 0.6, the analysis created a network with 415 out of the 420 genes, suggesting that these genes were all part of a stress response network (data not shown). When the Pearson’s correlation threshold was increased to |r| ≥ 0.8, a tight GCN consisting of 276 genes was obtained (Figure 4, Appendix A). This network was characterized by a nucleus of 112 highly co-expressed genes, with |r| ≥ 0.9 (Figure 4a). Several amongst these 112 co-expressed genes showed a high degree of connection and high betweenness centrality (Figure 4b, Appendix A). These genes represent the core of the analysed stress response network. 

This 112 genes core consisted of two well-defined modules of 38 and 72 genes (subgroups A and B, respectively; Figure 4c, Appendix A) within which all genes were highly positively correlated. Conversely, the subgroups A and B were strongly anti-correlated (Figure 4c). Two genes out of the 112 highly co-expressed genes, Os08g0492500 and Os09g0240200 (represented as two grey nodes in Figure 4c), did not belong to subgroups A and B, since they only showed negative correlations with some genes of subgroup B. 

A K-means cluster analysis that is based on the gene expression profiles of the 420 genes in the considered samples defined the presence of four different expression clusters (Appendix A). The optimum cluster number was determined based on converging results of the Sum of Squared Errors (SSE) estimate and the Calinsky criterion: for SSE, four clusters corresponded to the point where the SSE would not significantly decrease with each new addition of a cluster (first elbow); for Calinsky criterion, the maximum Calinski-Harabasz (CH) index was for two clusters, but also indices for three and four were still quite high and could be appropriate for the analysis (Appendix A). This analysis showed that all of the 38 genes of the subgroup A were part of the same expression cluster (Cluster 2; Appendix A) that was characterized by an up-regulation in the stress-treated samples. Differently, the 72 genes of the subgroup B belonged to two clusters (Clusters 1 and 3; Appendix A) characterized by a down-regulation in the stress-treated samples. Interestingly, Clusters 1 and 2 resulted in being strongly anti-correlated with (r) = −0.95 between the respective K-cluster centroids (Appendix A), thus confirming the anti-correlation between the subgroups A and B, which was observed in the Cytoscape graphical representation (Figure 4c). These data suggest that these subgroups act within the GCN in an antagonistic manner.

GO enrichment analysis of the two subgroups showed that they were characterized by different metabolic activities: transcription and regulation activities and hormone-related mechanisms (abscisic, jasmonic, and gibberellic acid) for the subgroup A; activities related to photosynthesis (chloroplast, thylakoid, stroma, chlorophyll), membrane/transmembrane related-mechanisms, and oxidation-reduction (oxidoreductase) reactions for the subgroup B (data not shown). 

Consistently, most of the 38 genes of the subgroup A are known to be involved in the response to abiotic stresses and plant hormones, in particular ABA and JA. Thirteen out of 38 genes (corresponding to 34%) encode TFs, most of which are known to be involved in stress response, in particular in ABA and JA-mediated signalling pathways (Figure 4c, Appendix A). These genes are described in details in the following sections. In addition, the gene *OsZFP15* (Os03g0820400), which encodes a zinc finger protein, has been related to stress response, although its mechanism of action is not well characterized. It was found to be differentially expressed between a cold-tolerant and a cold-sensitive rice genotype [27] and between a drought-tolerant and a drought-sensitive variety [38]. In the latter case, some polymorphisms were also observed between the genomic sequences of the two genotypes. More recently, this gene was related to drought-tolerance mechanisms, as it was found to be differentially regulated in WT and *erf71* transgenic rice plants under drought stress [50]. This gene is one of the putative hubs of the network, as it showed both a high degree of connection and betweenness centrality in the GCN (Figure 4b). Moreover, three genes belonging to the subgroup A encode kinases putatively involved in the transduction of the signalling cascade: OsRLCK253a and OsRLCK253b (Os08g0374600 and Os08g0374701, respectively) are two receptor-like cytoplasmic kinases that are probably involved in the salt stress response [51,52], while OsNPKL4 (Os01g0699600) is a MAPKKK that is strongly induced by drought and, to a lesser extent, by salt or cold stresses [53] (Figure 4c). These observations confirmed the putative upstream role of the subgroup A in the abiotic stress response pathway in rice. 

Differently, subgroup B showed a lower representation of TF encoding genes, which were only five out of 72 genes (7%). Three of them (Os03g0786400, Os11g0143300, and Os12g0139400) may have a key role in stress response, and they will be discussed below in detail. The TF-encoding gene Os10g0483000 (*Osj10gBTF3*) plays a role in seed germination and seedling growth and in pollen development [54,55]. Moreover, several genes of subgroup B are involved in photosynthesis or growth. In particular, 11 genes out of the 72 genes belonging to the subgroup B were related to photosynthesis (e.g., Os01g0279100, probably involved in chlorophyll biosynthetic processes, and the two genes Os07g0148900 and Os04g0635700, involved in photosystem assembly). It is well known that many cellular processes promoting plant growth and development are inhibited during the occurrence of environmental constraints, and photosynthesis and cell growth are among the primary processes to be affected by abiotic stresses [56]. Our hypothesis is that the subgroup B might principally represent those genes related to specific cell processes that are inhibited during the occurrence of an environmental cue to promptly activate stress response mechanisms.

Subgroups A and B also included genes of unknown function that may have a role in plant tolerance to different stresses (Appendix A). Interestingly, gene Os06g0133500 belonging to subgroup A was found to be up-regulated across different abiotic and biotic stresses [57], which suggests a role in multiple stress response. The gene Os03g0166000 in subgroup B encodes a member of the DNA-binding Alba (Acetylation Lowers Binding Affinity) protein superfamily, which might regulate gene expression through acetylation–deacetylation. Recently, the OsAlba transcript profiles under dehydration, hypersalinity, heat, cold, and phytohormone treatments indicated that most *OsAlba* genes might play a crucial role in stress adaptation [58]. A further gene belonging to the subgroup B, Os10g0475000, which showed a high degree and the highest betweenness centrality index among the 112 genes in the GCN, encodes a putative alcohol oxidase whose involvement in stress response has not been investigated so far (Figure 4b). 

Based on GCN analysis, we identified the signalling or metabolic pathways that were mainly represented in the GCN, thus with a prominent role in the differentiation between contrasting phenotypes in response to abiotic stresses.

### 2.5. ABA Synthesis and Metabolism

The abiotic stress response in plants is closely related to endogenous ABA levels and the regulation of ABA biosynthesis and metabolism is a key element for plant stress response [59]. A strict control of the expression of genes that are involved in ABA synthesis and catabolism is a critical point for a successful response of plant to abiotic stresses. According to the importance of this hormone, genes that are involved in ABA synthesis and catabolism were found amongst the 112 genes representing the GCN core (Figure 4c, Appendix A). 

In subgroup A, *OsABA8ox1* (Os02g0703600) codes for an ABA 8′-hydroxylase (ABA8ox) that catalyses the major regulatory step of the predominant pathway for ABA inactivation. This inactivation step involves the oxidation of ABA to 8′-hydroxy-ABA, which is later spontaneously isomerized to phaseic acid [60]. Previously, *OsABA8ox1* was shown to be up-regulated in roots under short-term osmotic stress in the tolerant cultivar Eurosis, but not in the susceptible Loto, and it was hypothesized that the encoded hydroxylase might contribute to the tolerant response of Eurosis through the reduction of ROS, which are produced during ABA signalling [28]. It was also demonstrated that the reduced ABA content in leaves of the salt-tolerant variety Baldo was related to tolerance and was linked to the down-regulation of *OsABA8ox1* [46]. Moreover, the overexpression of *OsABA8ox1* in rice highlighted that this gene is a master regulator of the abiotic stress response through the control of ABA level, whose fluctuation might vary the tolerance of rice to abiotic stresses [60]. Therefore, the adequate regulation of endogenous ABA levels, that is due, at least in part, to the activity of *OsABA8ox1*, is thought to be crucial for tolerance in rice [28,46,60]. 

In subgroup B, a further gene involved in ABA metabolism is present: Os04g0379700 encodes a violaxanthin de-epoxidase (VDE), which catalyses the conversion of violaxanthin to zeaxanthin through de-epoxidation. ABA biosynthesis starts from the epoxidation of zeaxanthin to violaxanthin, catalysed by zeaxanthin epoxidase, which is the forward step of VDE [61]. The equilibrium between VDE and zeaxanthin epoxidase might influence ABA biosynthesis during the occurrence of abiotic stresses, since a down-regulation of VDE might increase ABA biosynthesis. In addition, VDE is a key step of the xanthophyll cycle, which largely contributes to non-photochemical quenching (NPQ) to avoid photoinhibition, occurring when the light energy absorbed by plant leaves exceeds its consumption, through the dissipation of excessive light energy as heat [62]. Consistently, VDE is rate-limiting for NPQ under subsaturating light or during chilling in Arabidopsis [63] and overexpression in tomato of a *VDE* gene alleviates photoinhibition during high light and chilling stress [64]. Interestingly, we previously hypothesised that NPQ can be involved also in the response to salt stress, as we observed that salt-tolerant rice plants have an increased NPQ [32]. In the present study, the VDE-encoding gene Os04g0379700 showed the highest number of neighbours within the subgroup B and high betweenness centrality index of the GCN (Figure 4b), thus confirming its crucial role in the tolerance to abiotic stresses and supporting the hypothesis that VDE activity can be essential in this kind of response.

Our hypothesis is that the ability of rice genotypes to finely tune *OsABA8ox1* and *VDE* gene expression to reach the right balance between ABA biosynthesis, ABA catabolism, and NPQ mechanisms may be critical in tolerating the occurrence of adverse environmental cues.

### 2.6. The ABA-Mediated Response Pathway and the Crosstalk with Other Signalling Pathways

Some genes of the subgroup A are known to be involved in the ABA-dependent stress response pathway. *OsHsfA7* (Os01g0571300) encodes a Heat Shock Factor (HSF) that is involved in the rice response to ABA, heat, drought, and salt stresses, and it is able to increase tolerance to salt and drought stresses when overexpressed [65,66,67]. In our study, *OsHsfA7* expression was strongly and positively correlated with the transcription of two late embryogenesis abundant (LEA) proteins: Os08g0327700 and *OsRab16A* (Os11g0454300), with Pearson correlation values of 0.95 and 0.88, respectively. LEA proteins, and dehydrins in particular, play an important role as molecular chaperones in defending protein structures in plant cells [68]. *OsRab16A* encodes an ABA-inducible dehydrin, which is involved in the ABA-mediated response to abiotic stresses [67,69]. Its overexpression confers tolerance to salt and drought stresses [70,71]. It is noteworthy that *OsHfA7* was found to be co-expressed with 35 genes, including *OsRab16A*, in a gene module that is associated to drought response [65]. It was hypothesised that the TF OsHfA7 is able to recognize ABRE elements in these co-expressed genes and that this gene module is involved in the ABA-dependent pathway of drought response in different rice varieties [65]. Our data are in agreement with this study, thus confirming the important role of OsHfA7 in abiotic stress response.

Four *WRKY* genes are present in the subgroup A: *OsWRKY24* (Os01g0826400), *OsWRKY70* (Os05g0474800), *OsWRKY71* (Os02g0181300), and *OsWRKY108* (Os01g0821300; Figure 4c, Appendix A). These genes showed a strong and positive correlation with three genes coding for the TIFY/JA-ZIM domain (JAZ) TFs (Os03g0180800, Os03g0181100, and Os10g0391400) and with two genes coding for ERF TFs (Os04g0610400 and Os08g0474000). The JAZ and ERF TFs are involved in the JA and ethylene signalling pathways, respectively [72,73]. Consistently, these *WRKY* genes are involved in ABA-mediated signalling pathway and the cross-talk among pathways that are mediated by different hormones [74,75,76,77].

The TFs OsWRKY24, OsWRKY70 and the ERF OsAP2-39 (Os04g0610400) play a role in the same signalling pathways. OsWRKY24 and OsWRKY70 are repressors of the ABA and gibberellins (GA) signalling pathway [76,77]. Similarly, OsAP2-39 controls the ABA/GA balance, which in turn regulates plant growth and seed production by regulating both ABA biosynthesis and GA metabolism [78]. OsAP2-39 suppresses ethylene production, enhancing drought stress tolerance [79]. Moreover, *OsAP2-39* was found to be an important gene for drought tolerance in an agricultural environment, since it showed a significant Genotype x Environment interaction for drought [29]. Interestingly, a contrasting regulation of *OsAP2-39* expression between a tolerant and a sensitive genotype under severe drought stress has been reported [80]. Similarly, *OsWRKY70*, which is up-regulated under cold stress, is related to cold tolerance in three cold-tolerant genotypes [26]. Moreover, *OsWRKY24* and *OsWRKY71* are both involved in ABA-dependent pathways that are related to cold [74,81,82] and oxidative stress response [83], and are in network with *OsWRKY108* and a TIFY TF encoding gene, which is involved in JA, ethylene, and ABA signalling [75]. 

Our data are consistent with these previous studies and they confirm a possible role of these *WRKY* and *ERF* genes in the interconnections among different signalling pathways during abiotic stress response in rice. 

### 2.7. The JA-Mediated Response Pathway

Several genes belonging to the subgroup A are known to be involved in the JA signalling pathway. *OsCYP94C2a* (Os11g0151400) and *OsCYP94C2b* (Os12g0150200) encode cytochrome P450 proteins that are homologous to Arabidopsis *CYP94C1*, which hydroxylates JA-Ile, an active form of JA, and it is involved in inactivating the JA response [84]. *OsCYP94C2b* has been shown to repress the JA response and confer salinity tolerance in rice [84]. A similar role in the inactivation pathway of JA response might also be hypothesized for *OsCYP94C2a*. Interestingly, *OsCYP94C2a* and *OsCYP94C2b* were putative hub genes of the network, as they showed both a high degree of connection and betweenness centrality within the GCN (Figure 4b). The role of these genes in balancing the activation of the JA-mediated response pathway may be crucial for a correct response to abiotic stress. Indeed, JA signalling must be strictly controlled to avoid growth disadvantages, since it contributes to the production of defence-related proteins and metabolites, but it could also induce senescence through the degradation of chlorophyll and plastidial proteins [16,85]. 

Several TFs of the subgroup A are related to the JA-mediated response pathway. *OsZOS3-12* (Os03g0437200) is a JA-dependent C2H2-type zinc finger TF that is known to be involved in the salt stress response [52,86]. The genes Os03g0180800, Os03g0181100, and Os10g0391400, as mentioned above, code for the JAZ TFs OsJAZ9, OsJAZ10, and OsJAZ13, respectively [87]. As observed for *OsCYP94C2a* and *OsCYP94C2b*, the gene *OsJAZ13* might be considered a hub of the network for its values of degree and betweenness centrality indices of the GCN (Figure 4b). JAZ TFs act as repressors in the JA signalling pathway [72]. In rice, *OsJAZ9*, *OsJAZ10,* and *OsJAZ13* are OsMYC2-dependently induced by JA [88]. OsMYC2 is a bHLH TF that plays a key role in the negative feedback control of JA signalling in rice to avoid its negative effects [72]. *OsJAZ9*, *OsJAZ10,* and *OsJAZ13* are known to be involved in drought, salt, and cold stress response and *OsJAZ9* overexpression improves the salt and mannitol tolerance in rice plants [87]. It has been reported that the interaction of OsJAZ9 with other TFs plays a role in the response to abiotic stresses in rice. OsJAZ9 interacts with several bHLH TFs, with which it forms a transcriptional regulation complex to fine tune the expression of JA-responsive genes that are involved in salt stress tolerance in rice [89]. Recently, it was found that OsJAZ9 also interacts with OsMYB30 to negatively regulate the expression of *β*-amylase genes. This mechanism finely tunes the starch breakdown and the cell content of maltose, which might contribute as a compatible solute to cold tolerance in rice [90]. It is then possible that some genes of the subgroup A code for TFs that are able to interact with these JAZ proteins, thus influencing stress response. This hypothesis is strengthened by the presence of *OsbHLH148* (Os03g0741100) among the genes of the subgroup A. *OsbHLH148* is a bHLH gene whose expression level increases after treatment with methyl-JA or ABA, and under abiotic stresses, including dehydration, high salinity, low temperature, and wounding [91]. OsbHLH148 interacts with different OsJAZ proteins. In particular, the interaction of OsbHLH148 with OsJAZ1 is an important step of a JA signalling pathway, where ABA and JA act synergistically in response to stress, which leads to drought tolerance in rice [91]. In our data, the correlation of expression between *OsbHLH148* and the three *OsJAZ* genes was very high (Pearson correlation values of *OsbHLH148* with *OsJAZ9*, *OsJAZ10,* and *OsJAZ13* were 0.86, 0.86, and 0.9, respectively), which indicated a possible interaction between these TFs in a JA-mediated signalling of stress response. Our data strongly suggest that these TFs may have a prominent role in the response to abiotic stresses and a fine-tuning of their expression might influence the response phenotype (tolerance/susceptibility) of the rice genotypes.

It is noteworthy that several genes above described show a differential expression under cold stress between a tolerant and sensitive rice genotypes (*OsZFP15*, *OsWRKY24*, *OsWRKY70*, *OsWRKY71*, *OsAP2-39*, Os08g0474000, *OsZOS3-12*, *OsJAZ9*, *OsJAZ10*, *OsJAZ13*, and *OsbHLH148*) [27] and under salt stress at transcriptional and/or translational levels in two contrasting rice genotypes (*OsCYP94C2a*, *OsJAZ13, OsHsfA7*, *OsRab16A,* and *OsWRKY70*) [36]. These data support our hypothesis regarding a putative role of these genes in the differentiation of sensitive and tolerant phenotypes in response to abiotic stresses.

### 2.8. The DST-Related Pathway 

Amongst the TFs present in the subgroup B, Os03g0786400 showed both a high degree of connection and betweenness centrality within the GCN (Figure 4b), and it can be considered to be a hub gene of the network. Os03g0786400 codes for the zinc finger TF “Drought and Salt Tolerance” (DST), which is involved in the negative regulation of drought and salt tolerance in rice [92]. In particular, DST negatively regulates stomata closure by the direct modulation of genes related to homeostasis of H_2_O_2_, which mediates ABA-induced stomata closure [93]. The loss of DST function increases stomatal closure and reduces stomata density, consequently resulting in enhanced drought and salt tolerance in rice [92]. Our previous finding that salt-tolerant plants are able to close stomata more efficiently than salt-sensitive plants [32,46] is consistent with the putative upstream role of DST in the tolerance response. DST regulates the expression of genes that are related to H_2_O_2_ homeostasis, through the formation of a transcriptional complex with another TF, the “DST Co-activator 1” (DCA1) [94]. Among the genes that were regulated by DST, the peroxidase 24 precursor (*Prx24*, Os01g0378100) encodes an H_2_O_2_ scavenger highly expressed in guard cells [92,94]. Interestingly, *Prx24* is present among the 420 genes that were selected in this study, which suggests that the DCA1-DST-Prx24 pathway might have a role in the differentiation of the stress response phenotype among the analysed rice cultivars. 

The activity of DST is also linked to cytokinins (CK); indeed, it contributes to seed production via controlling CK degradation [95]. Interestingly, in our data, the expression of *DST* was strongly positively correlated with the two genes, *OsRR9* (Os11g0143300) and *OsRR10* (Os12g0139400; Pearson correlation values of 0.77 and 0.86, respectively), belonging to the subgroup B (Figure 4c, Appendix A). These genes code for A-type response regulators (RR), which are part of the CKs signalling cascade [96,97]. CKs have been implicated in plant development and stress response [98,99]. Our observation regarding these genes suggests that DST might be involved in a CK-mediated signalling pathway, where OsRR9 and OsRR10 have a signalling role, and this pathway is inhibited during the response to abiotic stresses. Consistently, *OsRR9*, which is down-regulated under cold stress, is related to cold tolerance in three cold-tolerant genotypes [26].

### 2.9. MYBs-Guided Subnetwork Interacts with Flavonoid Biosynthesis and ROS Response

MYB TFs are known to activate specific pathways and metabolite biosynthesis in response to abiotic stresses [100,101,102]. Three *MYB* genes were amongst the 276 genes of the GCN and two of them, *OsMYB55/61* (Os01g0285300) and *OsMYB61L* (Os05g0140100), were strongly correlated (*r* = 0.82). *OsMYB55/61* and *OsMYB61L* are transcriptionally regulated by the same TF, the NAC TF OsSND2 [103]; this might explain the strong correlation of their expression level. *OsMYB55/61* (with the name of O*sMYB2*) was reported to be one of the genes encoding TFs (SalTFs) localized within the major quantitative trait locus (QTL) for salinity tolerance “Saltol” [104]. Interestingly, *OsMYB55/61* is strongly and constitutively expressed in the salt-tolerant Pokkali genotype, but not in the sensitive cultivar IR64 [104]. Moreover, *OsMYB55/61* was shown to be involved in secondary cell wall biosynthesis by enhancing cellulose biosynthesis genes expression [103,105], while *OsMYB61L* was observed to be differentially expressed under salt stress in two contrasting rice genotypes [34]. *MYB* genes and their close neighbours (Pearson’s correlation value |r| ≥ 0.8) were enclosed in the core network to investigate the relationship between these *MYB* genes and the identified 112 genes of the stress response core (represented as yellow, beige, and violet nodes in Figure 4c; Appendix A). *MYB*s-guided subnetwork displayed positive correlation with some genes of the subgroup B and negative correlation with two genes belonging to the subgroup A, suggesting that it might act in frame with the subgroup B (Figure 4c, Appendix A). Consistently, the genes of this *MYBs*-guided subnetwork belonged to the expression Clusters 1 and 3, as it was observed for the genes of the subgroup B (Appendix A).

It is noteworthy that some genes in the *MYBs*-guided subnetwork code for enzymes that are involved in the flavonoid biosynthesis pathway (Os01g0106300, Os06g0102100, Os11g0530600; Figure 4c, Appendix A). Flavonoid biosynthesis is initiated by chalcone synthase (CHS), followed by chalcone isomerase, and the resulting flavanones are precursors of different classes of flavonoids, including flavones, flavonols, proanthocyanidins, and anthocyanins [106]. A meta-analysis showed that CHS is a drought-adaptive DEG that is shared across different species [40]. *OsCHS1* (Os11g0530600; also named *OsPKS26*) codes for a CHS, which was up-regulated in response to UV treatment in rice [107]. Interestingly, this gene was differently regulated under salt stress in a sensitive and a tolerant genotype. In particular, the induction of the flavonoid pathway appears to be a characteristic response to salt treatment of the sensitive genotype [108]. *OsCYP93G2* was a further gene related to flavonoid biosynthesis in *MYBs*-guided subnetwork (Os06g0102100), which codes for a flavanone 2-hydroxylase. This enzyme channels flavanones to biosynthesis of *C*-glycosylflavanones, a class of flavonoids with a role in the protection to UV light in rice [109,110]. It was shown that different allelic forms of *OsCYP93G2* in rice could cause the accumulation of different classes of flavonoids [111]. In addition, the gene Os01g0106300 that was present in the *MYBs*-guided subnetwork encodes a isoflavone reductase-like (IRL) protein, which is the enzyme responsible for production of medicarpin, a pterocarpan phytoalexin. Park et al. [107] suggested that the flavonoid biosynthetic pathway is closely related to the accumulation of phenolic phytoalexins in rice in response to UV treatment, in order to protect cells from oxidative stress. Consistently, the overexpression of another IRL gene, Os01g0106400, confers tolerance to ROS, which probably prevents an over production of ROS in rice cells [112]. Interestingly, Os01g0106300 is differentially expressed under salt stress in contrasting rice genotypes [33].

*OsCHS1* and *OsCYP93G2* can both be considered to be hub genes of the network, as they showed both high degree and betweenness centrality indices of the GCN (Figure 4b). These two genes belonged to both the *MYBs*-guided subnetwork and the subgroup B, where a phenylalanine ammonia-lyase (PAL) encoding gene (Os04g0518400) is also present (Figure 4c, Appendix A). PAL is the first enzyme of the phenylpropanoid pathway, from which the pathway of flavonoid synthesis branches off. The PAL gene showed a positive correlation with *OsMYB61L* (*r* = 0.76), which suggests that this MYB TF might act to change the equilibrium of the phenylpropanoid pathway, regulating flavonoid synthesis. These data suggested a crucial role for *OsMYB55/61* and *OsMYB61L* in controlling the expression of genes that are involved in flavonoid biosynthesis. A mechanism of stress response is the regulation of the equilibrium from flavonoid to lignin biosynthesis. Consistently, cold, drought, and salt stresses have been shown to alter lignin biosynthesis, with an impact on secondary cell wall formation and structure [113]. A fine tuning of this process to synthetize both lignin for cell wall formation and flavonoids for defence to ROS damage might ameliorate the stress tolerance of rice varieties. A natural variation of the expression of these genes occurs, as reported in some studies mentioned above [33,108,111], and it might differentiate the response to abiotic stresses of sensitive and tolerant rice genotypes. Consistently, a gene module related to flavonoid biosynthesis was identified as being associated with the abiotic stress response; the authors proposed that this module might play a role in abiotic stress tolerance by flavonoids as a ROS-scavenging system [43].

Moreover, *OsMYB61L* showed the highest positive correlation (*r* = 0.88) with Os01g0835500, named *OsNOX3*, which is involved in the response to ABA, drought, heat, and salt stress [36,114] (Figure 4c, Appendix A). *OsNOX3* codes for a NADPH oxidase-respiratory burst oxidase homologue (RBOH) protein [114]. Plasma membrane RBOHs play a key role in signal transduction reactions that mediate plant acclimation to abiotic stresses, since they are the primary sites of ROS production at the apoplast during abiotic stress condition [19]. We recently demonstrated a pivotal role for innate ROS scavenging systems and NOXs in fine tuning the intracellular H_2_O_2_ signalling and inducing salt-tolerance, both at the plant and single cell level [32,47,115]. Similarly, Saini and co-workers [35] observed a higher expression level of *OsNOX3* in the roots of a salt-tolerant cultivar than in those of a salt-sensitive genotype. The authors hypothesized a link between the higher expression of *OsNOX3* and the ability of the salt-tolerant cultivar to maintain higher H_2_O_2_ levels in comparison to the sensitive genotype. Our data suggested that *OsNOX3* might have a role in the abiotic stress tolerance of the genotypes analysed here. Recently, it was shown that ABA levels influenced *OsNOX3* expression [116]. ROS, and H_2_O_2_ in particular, are known to play several interactions with hormone mediated-signalling pathways and regulate both stomatal closure and the flavonoid biosynthesis pathway [20,21,93,112]. These data suggested that ROS fluctuations might have a role in the interconnection of the described pathways involved in the differentiation of tolerant and susceptible responses. Similar findings have been described in an integrated transcriptomic analysis in banana, which underlined a role of ABA and ROS-mediated signalling networks in the tolerance response to osmotic, cold, and salt stresses [117]. This suggests that some signalling pathways that are involved in tolerance to abiotic stresses are conserved among plant species. Moreover, our findings are consistent with the complex regulatory network, involving genes related to signal transduction, hormone-mediated signalling pathways, transcription regulation, and osmotic adjustment, observed in the abiotic stress-related meta-analysis that was recently performed by Smita and co-workers [43].

### 2.10. HUB Genes Distribution on Genome and Co-Localization with Stress-Related QTL 

Our analysis focused on those genes encoding for proteins with upstream roles in signal transduction (i.e., TFs and kinases) and on those genes that act in the same metabolic or signalling pathways, starting from the 112 highly co-expressed genes of the GCN. Indeed, we assumed that the over-represented pathways in this GCN probably play a primary role in tolerance response and in differentiation between contrasting phenotypes among genotypes. Most of the identified genes have been previously reported as being involved in the response to a single abiotic stress. Our analysis highlighted a role of these genes both (i) in the response to several abiotic stresses and (ii) in the differentiation between tolerant and susceptible responses. Moreover, three genes, whose function is not known, were considered as being hub genes of the network based on their position in the GCN or on a putative function in signal transduction and multiple stress response.

These 35 hub genes are listed in Table 4. 

A large number of QTLs that are related to tolerance to cold (CT), drought (DT) and salinity (ST) have been reported, and they are available on Q-TARO (QTL Annotation Rice Online) database [118]; therefore, we inspected whether the identified candidate genes were located within known QTLs. It is noteworthy that 24 out of the 35 candidate genes that were localized within at least one known QTL (Figure 5, Table 4, Appendix A). These genes represented 21 QTLs for DT, two for CT, and one for ST. The over representation of QTLs for DT reflects the higher number of studies related to drought stress traits that are reported in literature. Interestingly, four candidate genes co-localized with QTLs for tolerance to two stresses (i.e., *OsMYB55-61* and *OsHsfA7* for DT and ST, *OsABA8ox1* and *OsAP2-39* for CT and DT), consistently with their putative upstream role in rice tolerance to different abiotic stresses.

## 3. Materials and Methods

### 3.1. Transcriptome Data

Raw RNA reads were obtained from three studies analyzing the effect on transcriptomic profile in tolerant and susceptible rice seedlings for chilling stress [25], osmotic stress [28], and salt stress [32]. Although the three studies were independently conducted, their pipelines were similar. Briefly, *japonica* rice genotypes showing contrasting phenotypes (susceptible or tolerant) for the response to chilling, osmotic, or salt stresses had been separately selected for each stress condition. Afterwards, total RNA from seedlings of the contrasting genotypes grown under control and stress conditions had been extracted and sequenced using Illumina HiSeq2000 platform in three biological replicates. 

For the meta-analysis reported here, we used RNA reads data that were obtained from rice seedlings of contrasting genotypes grown under control conditions or treated with (i) 10 hs of chilling stress [25], (ii) 24 h of osmotic stress [28], and (iii) 72 h of salt stress [32], for a total of 36 samples. We opted for starting from the raw RNA reads in fastq format and carry out all the bioinformatics analyses from scratch using the most up-to-date software and databases in order to avoid the biases given by the different bioinformatic methods used in the original analyses. Table 1 summarizes the main features of the materials used for this study. 

A defined code was used to standardize the sample names: the first two letters of each sample name indicate the kind of treatment (Ch = chilling; Os = osmotic; Sa = salt), the following three letters are referred to the stress-related phenotype of the cultivar (Sus = susceptible; Tol = tolerant), the last letter indicates the growth condition (C = control; T = treated), and the final number represents the biological replicate (1, 2, or 3). 

### 3.2. RNA-Seq Data Handling and Mapping to Rice Genome

FastQC 0.11.7 [119] was used to assess the raw RNA reads quality of the thirty-six libraries, while trimmomatic 0.36 [120] was used to filter out the adaptors sequences and the low quality bases. The filtered RNA reads were then mapped to *Oryza sativa* ssp. *Japonica* (Nipponbare IRGSP-1.0) reference genome while using HiSat2 2.1.0 aligner [121] with default parameters. Finally, read counts were generated from alignment files with featureCounts software, part of Subread package 1.6.2 [122]. Namely, reads counting was carried out with featureCounts default parameters, basing on rice RAP-DB annotation gtf file version 1.0.38 and grouping ‘exon’ feature (‘-t exon’) into ‘gene_id’ meta-feature (‘-g gene_id’). Multi-mapping and multi-overlapping reads were not counted, and flag for paired-end reads (‘-p’) was added to the command line for chilling the experiment samples.

### 3.3. Differential Expression and GO-Enrichment Analyses

Differential expression analyses were separately carried out for the three stress experiments while using EdgeR 3.16.5 [123] on the 35,667 Nipponbare IRGSP-1.0 rice transcripts, annotated by RAP-DB, and downloaded from RAP-DB website (http://rapdb.dna.affrc.go.jp/) on 19 February 2018. EdgeR was used to: (i) filter out the not expressed or poorly expressed genes (we considered as “active” the genes with counts per million >1 in at least two libraries), (ii) normalize the RNA libraries, and (iii) do the differential expression analysis with the likelihood ratio test comparing treated (stressed) samples to control (not stressed) ones. Multi-dimensional scaling (MDS) plots for RNA libraries normalized counts were visualized for the three experiments while using the ‘plotMDS’ EdgeR command with default parameters. The Log_2_ Fold Change (LFC) of expression between treated and control samples was calculated with EdgeR, whose computing approach fits a negative binomial generalized linear model (GLM) to the read counts for each gene. The genes with a resulting false discovery rate (FDR) smaller than 0.05 were considered as DEGs. No LFC cut off was used for DEGs identification. The visualization of DEG intersections among the considered genotypes and treatments was performed while using UpSet R package [124]. Gene ontology (GO) enrichment of various DEGs subsets was characterized with AgriGO Singular Enrichment Analysis analytic tool [125] using default parameters (Fisher statistical test method; Yekutieli multi-test adjustment method; significance level of 0.05) and “Rice NCBI ID” as reference.

### 3.4. Correlation, Network and Clustering Analyses

RPKM data of genes involved in tolerant response to all of the three stresses were log-transformed while using log2(x + 1) for normalization, and Pearson pairwise correlation analysis was conducted across the selected samples using the “corrplot” and “hclust” R packages [126]. Significant correlations (*p* ≤ 0.05) with an absolute Pearson’s correlation coefficients |r| ≥ 0.8 were used for the construction of co-expression networks and network analysis in the Cytoscape software platform v. 3.5.1 [127]. 

Cluster analysis to identify similar transcript profile trends upon different stress conditions was carried out using K-means clustering in R following the 2-BitBio protocol (https://2-bitbio.com/2017/10/clustering-rnaseq-data-using-k-means.html). Mean RPKM expression values were log-transformed using log2(x + 1) followed by data scaling. To determine the optimum number of clusters, the methods of sum of squared error (SSE), the average silhouette width and the Calinski-Harabasz index, based on the intra- and inter-cluster sum of squares, were used [128,129,130].

## 4. Conclusions

In the present study, we performed a meta-analysis to identify the potential candidate genes for improving tolerance to several abiotic stresses in rice. We focused our attention on those genes whose stress-induced variation in their expression differed between the susceptible and tolerant genotypes under chilling, osmotic, and salt stresses. Gene network analysis recognized specific genes as major hubs of stress response (Table 4). Many of them encode TFs, which suggests that the differentiation between tolerance and susceptibility concerns upstream regulatory pathways. Some of these genes were previously described for their involvement in abiotic stress response. Literature data here reported, which pointed out the differences in their expression regulation between susceptible and tolerant genotypes, strengthen our hypothesis regarding their prominent role in the tolerance phenotype. As far as the signalling pathways that are involved in the differentiation of tolerant and susceptible response are concerned (Figure 6), we highlighted a prominent role of (i) the regulation of endogenous ABA levels during stress response through the modulation of genes related to ABA biosynthesis and catabolism, (ii) the ABA- and JA-mediated response pathways, (iii) the DST TF, involved in the negative regulation of drought and salt tolerance in rice via stomata aperture control, and (iv) MYB TFs in the regulation of flavonoid biosynthesis. An important role of ROS in the cross-talk among stress-related signalling pathways is hypothesized. 

In addition, three novel genes that may have an important role in the stress response network of rice genotypes were identified. Functional analyses of these genes are needed to understand their role in stress response.

A fine-tuning of the expression of the identified hub genes seems to be the key for the successful response to abiotic stresses of the tolerant rice genotypes. Further analyses are needed to understand their precise role in the response pathways and the interactions among the expressed proteins. Nevertheless, these genes have been recognized here as key players for conferring tolerance to chilling, osmotic, and salt stresses in rice. The identification of stress response hubs and the characterization of the molecular mechanisms behind the tolerance of rice genotypes might facilitate the generation of transgenic rice plants with enhanced abiotic stress tolerance, or the individuation of natural variants conferring higher tolerance to environmental constraints.

## Figures and Tables

**Figure 1 ijms-20-05662-f001:**
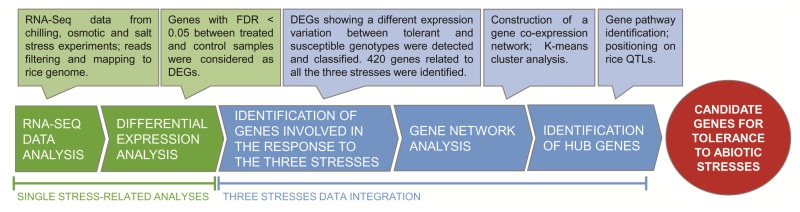
Overall work flow used in this study for the identification of candidate genes that are involved in tolerance to chilling, osmotic, and salt stresses in rice.

**Figure 2 ijms-20-05662-f002:**
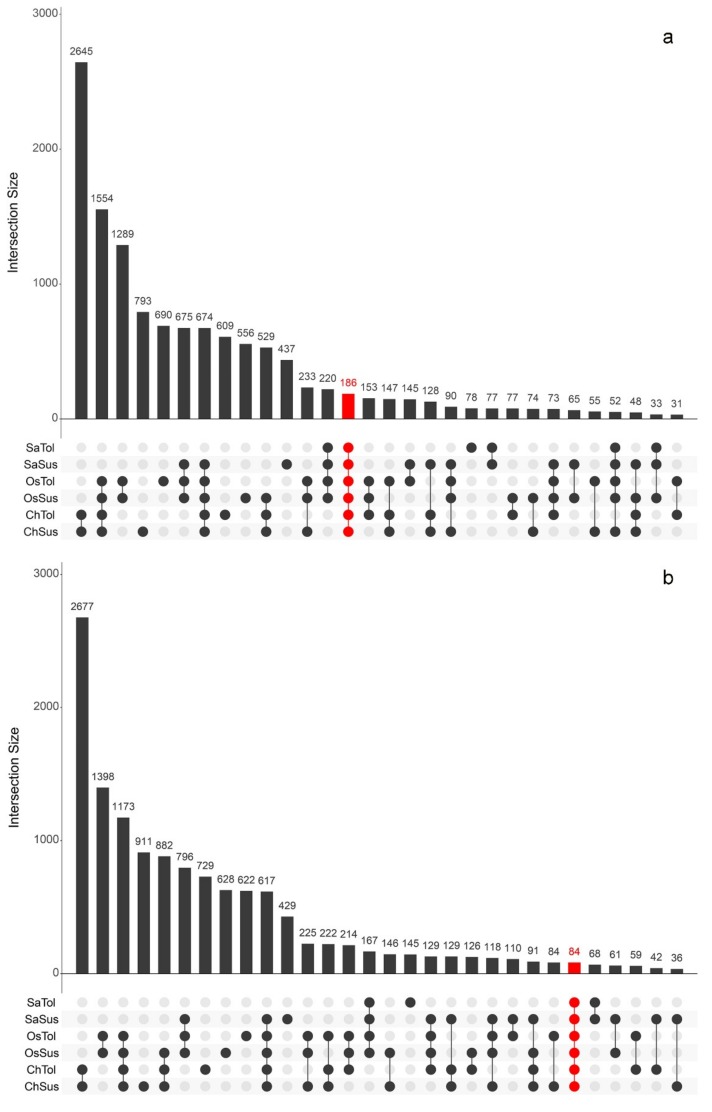
Intersecting DEGs sets for up-regulated (**a**) and down-regulated (**b**) genes among the six analysed genotypes. Only the 30 intersections with the highest number of DEGs were shown. The number of intersecting DEGs for each combination of group of genotypes was reported over the corresponding bar.

**Figure 3 ijms-20-05662-f003:**
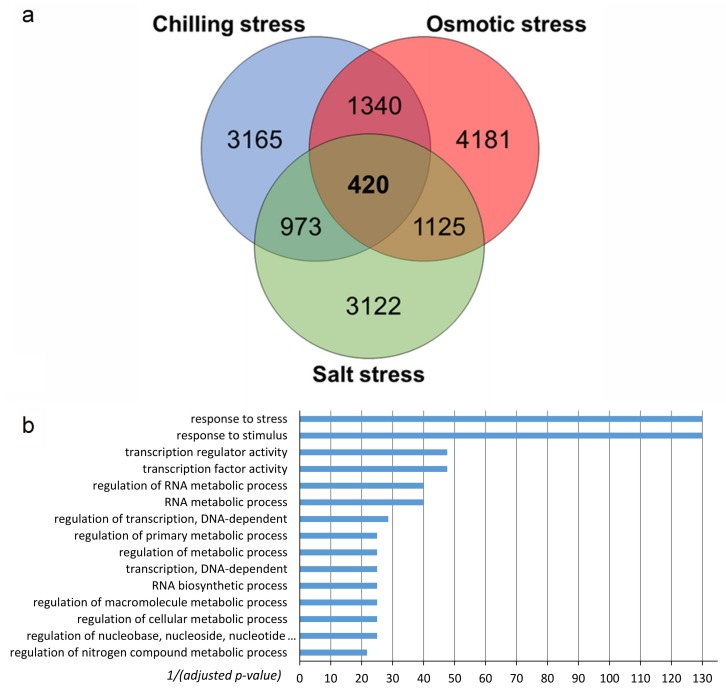
(**a**) Venn diagram of the genes which are differentially regulated in the susceptible or tolerant genotypes under chilling, osmotic or salt stress (**b**) GO-enrichment analysis results of 420 genes regulated in at least one cultivar in all of the three considered treatments.

**Figure 4 ijms-20-05662-f004:**
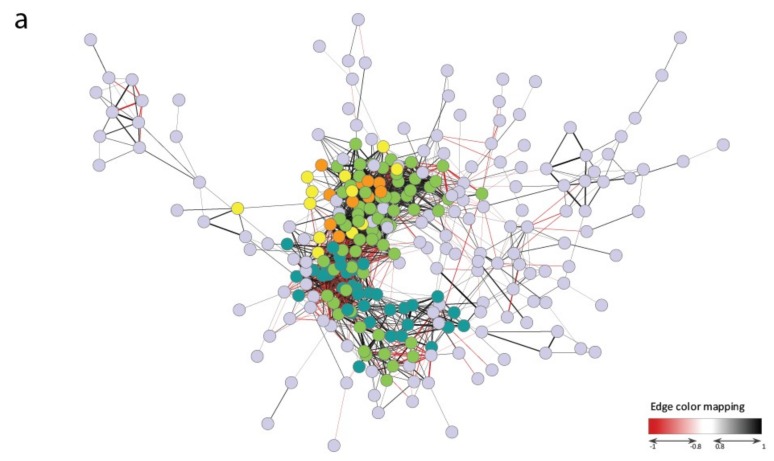
Gene Co-expression Network (GCN) created with Cytoscape. The nodes indicate the genes belonging to the network: Deep green, light green and yellow nodes represent genes belonging to subgroups A and B and to MYB subnetwork, respectively. Brown and orange nodes indicate genes in common between MYB subnetwork and subgroups A or B, respectively. Black and red edges indicate positive and negative correlations, respectively. The Cytoscape tool “Continuous mapping editor for edge width”, whose legend is shown in (**a**,**c**), was used to highlight the different strength of correlations, so that thickness of edges is proportional to correlation values. (**a**) Graphical representation of the GCN of 276 genes. (**b**) Relationship between the Number of Neighbours (Degree) and Betweenness Centrality of the 112 highly co-expressed genes. Light blue area contains those genes with both degree and Betweenness Centrality values greater than the 75th percentile of the relative distributions. For genes discussed in the text, ID or gene name is reported. Among them, genes encircled with deep green and light green belong to subgroups A and B, respectively; genes circled with orange belong to both subgroup B and MYB subnetwork. (**c**) Graphical representation of the hub signalling network (subgroup A and B) and of the MYB subnetwork. The discussed genes are represented as larger nodes; IDs or gene names are reported, TF encoding genes are represented as hexagon. The two grey nodes represent the genes Os08g0492500 and Os09g0240200, not belonging to neither subgroup A or B.

**Figure 5 ijms-20-05662-f005:**
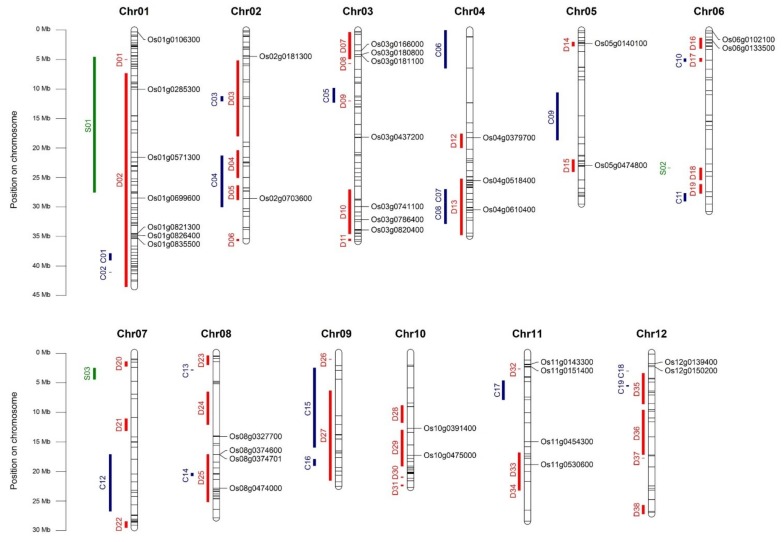
Distribution on rice chromosomes of the 420 selected genes and known QTLs related to cold/chilling (C), to drought (D), and to salt stress (S) tolerance. The position of the 420 genes on the genome is defined with black lines. The genes listed in Table 4 are indicated with the corresponding RAP IDs. Appendix A reports the correspondence between stress-related regions (C01-C17, D01-D38, S01-S03) and QTLs from QTARO database.

**Figure 6 ijms-20-05662-f006:**
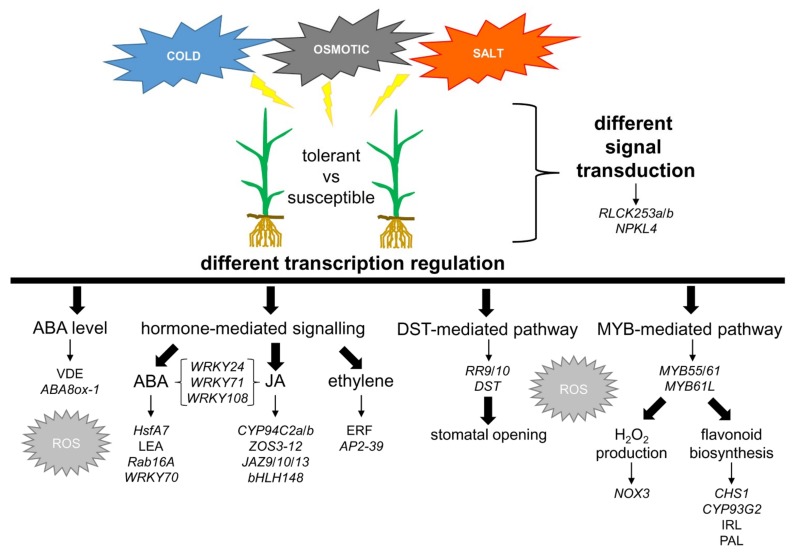
Schematic representation of genes and pathways putatively involved in the differentiation of tolerant and susceptible response to chilling, osmotic and salt stresses in rice.

**Table 1 ijms-20-05662-t001:** Description of RNA samples from [25,28,32] used for the meta-analysis. Stress conditions, sequencing techniques, cultivars and relative stress-related phenotype, and RNA sample names are listed.

Experiment	Sequencing Method	Cultivar	Condition	RNA Sample Names
Chilling stress (10 °C, 10 h)	75 bp paired-end	Thaibonnet chilling susceptible (ChSus)	Control	ChSusC1, ChSusC2, ChSusC3
			Treated	ChSusT1, ChSusT2, ChSusT3
		Volano chilling tolerant (ChTol)	Control	ChTolC1, ChTolC2, ChTolC3
			Treated	ChTolT1, ChTolT2, ChTolT3
Osmotic stress (PEG6000 20%, 24 h)	50 bp single-end	Loto osmotic stress susceptible (OsSus)	Control	OsSusC1, OsSusC2, OsSusC3
			Treated	OsSusT1, OsSusT2, OsSusT3
		Eurosis osmotic stress tolerant (OsTol)	Control	OsTolC1, OsTolC2, OsTolC3
			Treated	OsTolT1, OsTolT2, OsTolT3
Salt stress (saline solution [NaCl:MgSO_4_:CaCl_2_:NaNO_2_ = 10:2:1:1], 72 h)	50 bp single-end	Vialone Nano salt susceptible (SaSus)	Control	SaSusC1, SaSusC2, SaSusC3
			Treated	SaSusT1, SaSusT2, SaSusT3
		Baldo salt tolerant (SaTol)	Control	SaTolC1, SaTolC2, SaTolC3
			Treated	SaTolT1, SaTolT2, SaTolT3

**Table 2 ijms-20-05662-t002:** Statistics for differential expression analyses. For each experiment and cultivar, the number of active genes (showing read count per million bases >in at least two libraries) and differentially expressed genes (DEGs, FDR < 0.05) are reported. DEGs were also classified as up-regulated (logFC > 0) and down-regulated (logFC < 0) in treated compared to control samples.

Cultivar Phenotype	Chilling Stress		Osmotic Stress		Salt Stress	
	Susceptible	Tolerant	Susceptible	Tolerant	Susceptible	Tolerant
Cultivar name	ChSus	ChTol	OsSus	OsTol	SaSus	SaTol
Active genes	20,907	21,137	20,267	20,265	20,851	20,719
Number of DEGs (FDR < 0.05)	14,944	14,355	13,722	12,654	6197	1537
Up-regulated (logFC > 0)	7382	7039	6692	6416	3116	841
Down-regulated (logFC < 0)	7562	7316	7030	6238	3081	696

**Table 3 ijms-20-05662-t003:** Number of genes which were differentially regulated between susceptible or tolerant genotypes in response to each stress, according to their classification in the six classes described in the text: (i) up-regulated only in the susceptible genotype (“SusOnly_up”), (ii) down-regulated only in the susceptible genotype (“SusOnly_down”), (iii) up-regulated only in the tolerant genotype (“TolOnly_up”), (iv) down-regulated only in the tolerant genotype (“TolOnly_down”), (v) DEGs in both genotypes, showing a difference of log_2_FC between tolerant and susceptible cultivars higher than 1 (“ΔLFC > 1”), and (vi) DEGs in both genotypes, showing a difference of log_2_FC between tolerant and susceptible cultivars less than −1 (“ΔLFC < −1”).

DEG Class	Chilling	Osmotic	Salt
SusOnly_up	1321	1355	2446
SusOnly_down	1556	1927	2639
TolOnly_up	964	1083	171
TolOnly_down	1324	1131	254
ΔLFC > 1	425	798	55
ΔLFC < −1	308	772	75
TOTAL	5898	7066	5640

**Table 4 ijms-20-05662-t004:** List of the potential candidate genes for tolerance to abiotic stresses in rice. Gene name and RAP ID are indicated. GCN: Gene Co-expression Network; CT: cold tolerance; DT: drought tolerance; ST: salt tolerance. Literature data showing a variation in gene expression levels between contrasting genotypes under a specific stress condition are reported. Co-localized QTL name (in italic) or trait description are shown.

Gene Name	RAP ID	Role in Stress Response	GCN Subgroup	References about Variations in Gene Expression between Contrasting Genotypes	Co-Localization with Known QTLs
*OsZFP15*	Os03g0820400	unknown function (TF)	A	cold [27] drought [38]	1 DT (panicle length)
*OsRLCK253a*	Os08g0374600	signal transduction	A	/	1 DT (osmotic adjustment)
*OsRLCK253b*	Os08g0374701	signal transduction	A	/	1 DT (osmotic adjustment)
*OsNPKL4*	Os01g0699600	signal transduction	A	/	1 DT: *qLRC-1*
*OsABA8ox1*	Os02g0703600	ABA catabolism	A	osmotic [28]	1 CT: *qSDW2*; 2 DT: *qGY-2b*, *qTGW-2a*
*VDE*	Os04g0379700	ABA biosynthesis/xanthophyll cycle	B	/	1 DT (panicle length)
*OsHsfA7*	Os01g0571300	ABA signalling (TF)	A	salt [36]	1 DT: *rfw1b*; 1 ST (Na^+^ uptake)
*LEA*	Os08g0327700	ABA signalling	A	/	/
*OsRab16A*	Os11g0454300	ABA signalling	A	salt [36]	/
*OsWRKY24*	Os01g0826400	ABA, GA, JA signalling (TF)	A	cold [27]	1 DT (Panicles/m^2^)
*OsWRKY70*	Os05g0474800	ABA and GA signalling	A	cold [27] salt [36]	2 DT (panicle or tiller no.per m^2^, fraction sterile panicles)
*OsWRKY71*	Os02g0181300	ABA, GA, JA signalling (TF)	A	cold [27]	/
*OsWRKY108*	Os01g0821300	ABA, JA signalling (TF)	A	/	1 DT (Panicles/m^2^)
*OsAP2-39*	Os04g0610400	ethylene signalling (TF)	A	cold [27] drought [80]	1 CT: *OsAOX1a*; 1 DT: *rfw4a*
*ERF*	Os08g0474000	ethylene signalling? (TF)	A	cold [27]	1 DT (osmotic adjustment)
*OsCYP94C2a*	Os11g0151400	JA inactivation	A	salt [36]	/
*OsCYP94C2b*	Os12g0150200	JA inactivation	A	/	/
*OsZOS3-12*	Os03g0437200	JA signalling (TF)	A	cold [27]	/
*OsJAZ9*	Os03g0180800	JA signalling (TF)	A	cold [27]	1 DT: *qtl3.1*
*OsJAZ10*	Os03g0181100	JA signalling (TF)	A	cold [27]	1 DT: *qtl3.1*
*OsJAZ13*	Os10g0391400	JA signalling (TF)	A	cold [27] salt [36]	/
*OsbHLH148*	Os03g0741100	JA signalling (TF)	A	cold [27]	2 DT (grains per panicle, carbon isotope discrimination)
*DST*	Os03g0786400	H_2_O_2_/CK signalling (TF)	B	/	1 DT (panicle length)
*OsRR9*	Os11g0143300	CK signalling (TF)	B	/	/
*OsRR10*	Os12g0139400	CK signalling (TF)	B	/	/
*OsMYB55-61*	Os01g0285300	TF	MYB subnet	salt [104]	1 DT: *rfw1b*; 1 ST (Na^+^ uptake)
*OsMYB61L*	Os05g0140100	TF	MYB subnet	salt [34]	1 DT (Sterility (%))
*OsCHS1*	Os11g0530600	flavonoid biosynthesis	MYB subnet/B	salt [108]	3 DT: *gpl11.1*, *gw11.1*, *yld11.1*
*OsCYP93G2*	Os06g0102100	flavonoid biosynthesis	MYB subnet/B	/	/
*IRL*	Os01g0106300	flavonoid biosynthesis	MYB subnet	salt [33]	/
*PAL*	Os04g0518400	phenylpropanoid/flavonoid biosynthesis	B	/	1 DT: *rfw4a*
*OsNOX3*	Os01g0835500	H_2_O_2_ signalling	MYB subnet	salt [35]	1 DT (Panicles/m^2^)
-	Os06g0133500	unknown function	A	/	1 DT (leaf rolling score)
-	Os03g0166000	unknown function (TF?)	B	/	2 DT: *rn3*, *qtl3.1*
-	Os10g0475000	unknown function (alcohol oxidase?)	B	/	1 DT (Root penetration index)

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
