# Peer review of "A Meta-Analysis of Comparative Transcriptomic Data Reveals a Set of Key Genes Involved in the Tolerance to Abiotic Stresses in Rice"

_ijms, 2019, doi:10.3390/ijms20225662_

Round 1
Reviewer 1 Report
In this manuscript the authors identified a cohort of rice genes responsive to abiotic stress through a meta-analysis of three published transcriptomic experiments contrasting susceptible and tolerant rice genotypes to individual abiotic stresses. Overall, the authors did a very thorough examination of their target gene set. However, I feel the methods used in their analyses could use some clarification, and the analysis / data associated with Figure 2 is confusing.
Regarding the methods specifically. First, the parameters used for featureCounts should be included. Were overlapping or multimapping reads included? What features / metafeatures were used for the expression counting? Was the paired-end flag used for the chilling experiment? Second, the parameters used for the AgriGO analysis should also be included as there are multiple statistical test methods and multiple-test adjustment methods which can be used. Further, the authors reported p-values for the GO enrichment, but they really should use multiple-test adjusted p-values (Fig 3b and Fig S3). Third, lines 158-160 list multiple approaches for determining the optimum number of clusters for k-means, but each often yield very different results. Which method was actually used to determine K = 4? In addition, the MDS analysis in Results and Discussion did not have an associated method description: What program / command was used for the MDS analysis and with what input data (raw counts / normalized counts)?
Regarding Figure 2, the values shown in the figure don’t match up with data elsewhere in the manuscript (particularly with Table 2). Based on how I am interpreting the analysis, Figure 2 should be a breakdown of the DEGs summarized in Table 2. However, the sum of the DEGs associated with each treatment / genotype in Figure 2 are always less than the summary given in Table 2. For example, SaTol and SaSus have a total of 648 and 2903 Up-regulated DEGs based on Figure 2, respectively, but Table 2 shows 841 and 3116. The differences seem to occur in all cases. If there was another filtering step included between the genes shown in Table 2 and the presentation in Figure 2, that needs to be stated. Otherwise Figure 2 needs to be corrected in order to match the totals in Table 2.
Additional specific comments:
Line 88: “this approach allows to obtain” is poorly worded.
Line 97: “The resulted data were then” should be “The resulting data…”
Line 124: “To uniform the names of these samples” is poorly worded.
Line 140: “genes with counts per million bases > 1” should be “…counts per million > 1”
Line 251: “1,398” should be “1,173.” The 1,398 is the total for OsTol and OsSus, not all Os and Ch.
Lines 263, 278: “cultivars minor than -1” should be “cultivars less than -1”
Line 501: “amilase” should be “amylase”
Line 606: “Several observations above reported [..] indicate” is poorly worded.
Fig 3b and S3: The graph really should have lines originating from the x-axis as it is difficult to align the bars to p-values
Figure 4a: Why are there a few black edges with a majority being grey?
Figure 4b: -The legend doesn’t explain what the colored circles around select points means.
-The text on the axis is very small and difficult to read, consider enlarging the text and
-reducing the frequency of the labels.
Figure 4c: Several items in this figure are not easy to distinguish:
-The violet vs grey nodes are very similar, try selecting more unique colors.
-The yellow vs beige nodes are very similar, try selecting more unique colors.
-The edge thickness being proportional to correlation is nearly impossible to discern.
Author Response
Point-by-point response to Reviewer 1
In this manuscript the authors identified a cohort of rice genes responsive to abiotic stress through a meta-analysis of three published transcriptomic experiments contrasting susceptible and tolerant rice genotypes to individual abiotic stresses. Overall, the authors did a very thorough examination of their target gene set. However, I feel the methods used in their analyses could use some clarification, and the analysis / data associated with Figure 2 is confusing. Regarding the methods specifically.
First, the parameters used for featureCounts should be included. Were overlapping or multimapping reads included? What features / metafeatures were used for the expression counting? Was the paired-end flag used for the chilling experiment?
Method description for featureCounts operations was added to Materials and Methods section (lines 135-138).
Second, the parameters used for the AgriGO analysis should also be included as there are multiple statistical test methods and multiple-test adjustment methods which can be used.
Method description for AgriGO operations was added to Materials and Methods section (lines 155-157).
Further, the authors reported p-values for the GO enrichment, but they really should use multiple-test adjusted p-values (Fig 3b and Fig S3).
In Fig. 3b and Fig. S3, we changed the p-value-based graphs with graphs based on multiple-test adjusted p-values. Moreover, according to another comment by Reviewer 1, vertical lines originating from x-axis were added.
Third, lines 158-160 list multiple approaches for determining the optimum number of clusters for k-means, but each often yield very different results. Which method was actually used to determine K = 4?
We have explained the strategy used to determine K = 4 in the text (lines 364-369) and we added Fig. S5 referred to this.
In addition, the MDS analysis in Results and Discussion did not have an associated method description: What program / command was used for the MDS analysis and with what input data (raw counts / normalized counts)?
Method description for MDS analysis was added to Materials and Methods section (lines 146-148).
Regarding Figure 2, the values shown in the figure don’t match up with data elsewhere in the manuscript (particularly with Table 2). Based on how I am interpreting the analysis, Figure 2 should be a breakdown of the DEGs summarized in Table 2. However, the sum of the DEGs associated with each treatment / genotype in Figure 2 are always less than the summary given in Table 2. For example, SaTol and SaSus have a total of 648 and 2903 Up-regulated DEGs based on Figure 2, respectively, but Table 2 shows 841 and 3116. The differences seem to occur in all cases. If there was another filtering step included between the genes shown in Table 2 and the presentation in Figure 2, that needs to be stated. Otherwise Figure 2 needs to be corrected in order to match the totals in Table 2.
In Fig. 2 not all of the intersections among samples are shown. Among the 61 possible intersections between the 6 groups of DEGs, we chose to report in Fig. 2 only the 30 categories including the highest numbers of genes, otherwise the figure would not have been readable (see the figures attached in the PDF file). We thank the Reviewer 1 for the relevant observation, as this information was not reported in any part of the manuscript. We reported the right information at lines 152-153, 241-246 and in Figure 2 caption (lines 249-250).
Additional specific comments:
- Line 88: “this approach allows to obtain” is poorly worded.
We revised the text (line 88).
- Line 97: “The resulted data were then” should be “The resulting data…”
We corrected as suggested (line 97).
- Line 124: “To uniform the names of these samples” is poorly worded.
We revised the text (lines 124-125).
- Line 140: “genes with counts per million bases > 1” should be “…counts per million > 1”
We changed as suggested (line 144).
- Line 251: “1,398” should be “1,173.” The 1,398 is the total for OsTol and OsSus, not all Os and Ch.
We corrected the mistake (line 264).
- Lines 263, 278: “cultivars minor than -1” should be “cultivars less than -1”
We changed as suggested (lines 276, 291).
- Line 501: “amilase” should be “amylase”
We corrected the mistake (line 539).
- Line 606: “Several observations above reported [..] indicate” is poorly worded.
We changed the sentence: “A natural variation of the expression of these genes occurs, as reported in some studies above mentioned” (lines 643-645).
- Fig 3b and S3: The graph really should have lines originating from the x-axis as it is difficult to align the bars to p-values
Vertical grey lines originating from x-axis were added to Fig. 3b and Fig. S3.
- Figure 4a: Why are there a few black edges with a majority being grey?
In this Cytoscape GCN, positive correlations are coloured in black whereas negative ones in red. We used the Cytoscape “Continuous mapping editor for edge width” tool to highlight the different strength of correlations. The scale legend and explanation of that was added to Figure 4a.
- Figure 4b: The legend doesn’t explain what the colored circles around select points means.
Coloured circles around selected point represent the same categories as in Figure 4a and c. This information was added to Figure 4 legend (lines 354-356).
- Figure 4b: The text on the axis is very small and difficult to read, consider enlarging the text and reducing the frequency of the labels.
The Figure 4 was changed consistently with all the reviewer suggestions. The character font of values and labels in Figure 4b was increased and the figure definition was improved in order to easily read values and labels.
- Figure 4c: - The violet vs grey nodes are very similar, try selecting more unique colors.
-The yellow vs beige nodes are very similar, try selecting more unique colors.
-The edge thickness being proportional to correlation is nearly impossible to discern.
Figure 4c was changed according to suggestions.
The legend of Figure 4 was modified according to these changes (lines 343-362).

Reviewer 2 Report
The authors carried out a meta-analysis of already published trascriptomic data with the purpose of identifying, through GCN analysis, key genes simultaneously involved in the response of rice to three abiotic stresses: cold, osmotic and salt stress. The experimental design is challenging as, in the well described set of samples, the susceptible and tolerant varieties were not the same in the three tested treatment and the RNA seq data were gained in 2 different ways (75 long paired ends and 50 long single reads). Nonetheless, as every comparison was made first between treated and untreated samples of the same genotype by computing logFC, I agree that every bias due to the mentioned factors can be considered as sufficiently neutralized. Besides, statement in lines 173-175 is coherent and compatible with my knowledge.
The authors evidence 420 genes involved somehow in the response to all the three stress conditions tested and, by GCN, they select 112 genes strictly connected to each other. Among these evidenced genes, they focus the attention on 35 genes (listed in tab4). Lines 357-634 are substantially dedicated to comments about the expression of these 35 genes as described in cited literature and to comparison with what was revealed in the present meta-analyses, substantially conforming previous findings and somewhere suggesting a more precise interpretation.
It is not completely clear the criterion followed in the selection of these 35 genes: in fact, 32 are already known as involved in abiotic stress response as belonging to known pathways (ABA, JA and so on), other three are genes of unknown function but were previously already pointed out as candidate in abiotic stress resistance since they mapped on well studied QTLs. The author should absolutely make an effort in better explaining and supporting the choice of discussing exclusively those genes. In fact, the major objective of meta-analysis of big data is pointing out novel genes involved in the network and not yet investigated.
I also add two minor tips:
computing LogCF by data in tab S2, doesn't give data in tabS3, author should explain if other normalization steps occurred and should preferably include in tab S2 definitive normalized data
please, verify ref citing in lines 533, 556 and 618
Author Response
Point-by-point response to Reviewer 2
The authors carried out a meta-analysis of already published transcriptomic data with the purpose of identifying, through GCN analysis, key genes simultaneously involved in the response of rice to three abiotic stresses: cold, osmotic and salt stress. The experimental design is challenging as, in the well described set of samples, the susceptible and tolerant varieties were not the same in the three tested treatment and the RNA seq data were gained in 2 different ways (75 long paired ends and 50 long single reads). Nonetheless, as every comparison was made first between treated and untreated samples of the same genotype by computing logFC, I agree that every bias due to the mentioned factors can be considered as sufficiently neutralized. Besides, statement in lines 173-175 is coherent and compatible with my knowledge.
The authors evidence 420 genes involved somehow in the response to all the three stress conditions tested and, by GCN, they select 112 genes strictly connected to each other. Among these evidenced genes, they focus the attention on 35 genes (listed in tab4). Lines 357-634 are substantially dedicated to comments about the expression of these 35 genes as described in cited literature and to comparison with what was revealed in the present meta-analyses, substantially conforming previous findings and somewhere suggesting a more precise interpretation.
It is not completely clear the criterion followed in the selection of these 35 genes: in fact, 32 are already known as involved in abiotic stress response as belonging to known pathways (ABA, JA and so on), other three are genes of unknown function but were previously already pointed out as candidate in abiotic stress resistance since they mapped on well studied QTLs. The author should absolutely make an effort in better explaining and supporting the choice of discussing exclusively those genes. In fact, the major objective of meta-analysis of big data is pointing out novel genes involved in the network and not yet investigated.
In this paper, we principally focused our interest on i) transcription factor-encoding genes, which are known to be key players in the regulation of stress response, and ii) genes involved in signalling or metabolic pathways over-represented in the analysed GCN, as they may likely have a primary role in tolerance response.
Most of the identified genes were previously shown as involved in the response to a single abiotic stress. Differently, our analysis highlighted a role of these genes both i) in the response to a range of abiotic stresses and ii) in the differentiation between tolerant and susceptible response. This is the added value of our study, which makes these genes excellent candidates for breeding programs or transgenic/genome editing approaches to achieve tolerance to different stresses. We better explained the rationale of our choice at lines 429-431 and 675-684.
Moreover, three genes of unknown function were identified as hub genes in the GCN network, and may play a major role in tolerance to different stresses. In particular:
- Os06g0133500 was found to be up-regulated across different abiotic and biotic stresses, as reported by Narsai et al. 2013 (reference n. 69)
- Os03g0166000 belongs to the DNA-binding Alba (Acetylation Lowers Binding Affinity) superfamily. ALBA proteins may regulate gene expression through acetylation–deacetylation and OsAlba transcript profiles indicated that most OsAlba genes might play a crucial role in stress adaptation (Verma et al 2018, new reference n. 70)
- Os10g0475000 is one of the genes with the highest degree and betweenness centrality indices in the GCN, and encodes an alcohol oxidase whose involvement in stress response has never been investigated (we corrected the wrong reference to Fig. 4c with the correct reference to Fig. 4b at line 428).
We better explained the importance of these genes at lines 414-428 and 733-735.
I also add two minor tips:
- computing LogCF by data in tab S2, doesn't give data in tabS3, author should explain if other normalization steps occurred and should preferably include in tab S2 definitive normalized data
No further normalization steps occurred on the Table S2 data. Actually, EdgeR software does not use a typical formula “log[(treated normalized count mean) / (control normalized count mean)]”, but uses a generalized linear model (GLM) approach which specifies the probability distributions according to their mean-variance relationship. In particular, EdgeR uses the value from the ‘prior.count’ parameter in ‘glmFit’ command, causing the returned coefficients to be shrunk, as reported in manuals
(EdgeR: https://bioconductor.org/packages/release/bioc/manuals/edgeR/man/edgeR.pdf
https://www.rdocumentation.org/packages/edgeR/versions/3.14.0/topics/glmFit).
To clarify this methodology, we mentioned the EdgeR GLM approach in Materials and Methods section (lines 149-150).
Anyway, to double check our results, we run EdgeR again from scratch, and we can confirm that the data on Tables S2 and S3 had been reported correctly.
- please, verify ref citing in lines 533, 556 and 618
We integrated these references in the numbering: “Formentin et al. 2018a, b” was modified with “[32,58]” (line 562), Ye et al. 2018 was numbered as [115], instead of [117] (and, consequently, 115 and 116 have been turned into 116 and 117, respectively; lines 585, 587, 589 and 591 and Table 4), and “Saini and co-workers (2018)” was replaced by “Saini and co-workers [35]” (lines 648-649). The numbering was also corrected in the reference list (lines 1022-1036).

Round 2
Reviewer 2 Report
The authors responded to my comments point by point and didn't neglect any suggestion.
I think that the presentation of results was significantly improved by a major emphasis on novel genes pointed out and I suggest that the paper can be now accepted for publication.